# Experimental evidence supporting a global melt layer at the base of the Earth's upper mantle

D. Freitas [1], G. Manthilake [1], F. Schiavi[1], J. Chantel[2], N. Bolfan-Casanova[1], M.A. Bouhifd[1] & D. Andrault[1]

The low-velocity layer (LVL) atop the 410-km discontinuity has been widely attributed to dehydration melting. In this study, we experimentally reproduced the wadsleyite-to-olivine phase transformation in the upwelling mantle across the 410-km discontinuity and investigated in situ the sound wave velocity during partial melting of hydrous peridotite. Our seismic velocity model indicates that the globally observed negative Vs anomaly (−4%) can be explained by a 0.7% melt fraction in peridotite at the base of the upper mantle. The produced melt is richer in FeO (~33 wt.%) and $H_2O$ (~16.5 wt.%) and its density is determined to be 3.56–3.74 g cm$^{-3}$. The water content of this gravitationally stable melt in the LVL corresponds to a total water content in the mantle transition zone of 0.22 ± 0.02 wt.%. Such values agree with estimations based on magneto-telluric observations.

[1] Université Clermont Auvergne, CNRS, IRD, OPGC, Laboratoire Magmas et Volcans, F-63000 Clermont-Ferrand, France. [2] Department of Earth, Environmental and Planetary Sciences, Case Western Reserve University, Cleveland, OH 44106, USA. Correspondence and requests for materials should be addressed to G.M. (email: geeth.manthilake@uca.fr)

The origin of the low velocity layer (LVL) atop the 410-km discontinuity has long been attributed to the presence of melt[1–6]. The LVL is reported to be a widespread seismic anomaly[5], with an average 4% Vs velocity drop across a narrow, ~50–60-km-thick region atop the mantle transition zone (MTZ). The presence of a partially molten layer at the base of the upper mantle has a wide range of geological implications. In particular, it is directly linked to $H_2O$ contents in the MTZ, which plays an important role controlling the global water circulation[7,8]. It promotes the material exchange between the upper and lower mantle without requiring a whole mantle or layered convection model and could help to explain the chemical differences between mid-ocean-ridge and ocean-island basalts[1]. Refining the LVL properties may provide unique insight into the nature of deep mantle plumes and their interactions with the surrounding mantle[1,6]. Further, a melt layer may influence mantle dynamics by decoupling the upper mantle and the MTZ above the 410 km discontinuity along this low-viscosity layer (LVL) and allowing coherent motion between lithosphere and the underlying mantle[6].

Numerous studies predict the melting of peridotite under deep mantle conditions if water is present[1,9,10]. The magnitude of the Vs velocity drop at the LVL is directly related to the melt volume fraction. The melt fraction in the LVL should be the key parameter controlling the stability of the melt layer and the subsequent element fractionation between upwelling mantle and the melt. The highly wetting character of hydrous silicate melt at high pressure[11] implies that even a small degree of melting can dramatically affect the propagation of seismic waves at high depths. Both experimental[12–14] and theoretical models[15–22] have investigated the magnitude of the velocity drop that would be associated with the melt fraction. However, the current velocity models are based on either low pressure (<2.5 GPa) laboratory measurements or simulations based on simplified melt geometries, limiting their applications when interpreting the seismic anomaly at the base of the upper mantle. Thus, systematic laboratory measurements of the effect on seismic velocity of a melt generated at deep mantle conditions are central to establishing the dehydration-melting scenario at the base of the upper mantle.

Density is a key parameter controlling the stability of a melt layer in the LVL[1,23–25], and is determined by chemical composition as well as pressure and temperature. However, experimental investigations of the hydrous peridotite melt composition are limited to 6 GPa[26]. Other systematic works have explored the density of hydrous silicate melts at high pressure, assuming variable concentrations of FeO and $H_2O$[10,27–33]. While these studies provide useful relations between melt density and composition, knowledge of the melt composition as a function of the degree of melting is currently lacking, adding notable uncertainties to current geodynamic models[23].

In this study, we experimentally simulate upwelling of peridotite mantle across the 410 km discontinuity by synthesizing variably hydrated peridotites in the wadsleyite stability field (14 GPa) and measuring the sound wave velocity of re-equilibrated peridotite in situ in the olivine stability field (12 GPa) as a function of increasing temperature. Then, using ex situ characterization of the melt chemical composition and microstructures, together with numerical calculations, we constrain the partial melting scenario during peridotite dehydration above the MTZ. Using sound wave velocity, we determined the melt volume fraction and, from chemical systematics, the composition of the melt that would be consistent with seismic observations. Finally, we discuss the geophysical and geochemical consequences of a globally distributed melt layer at the base of the Earth's upper mantle.

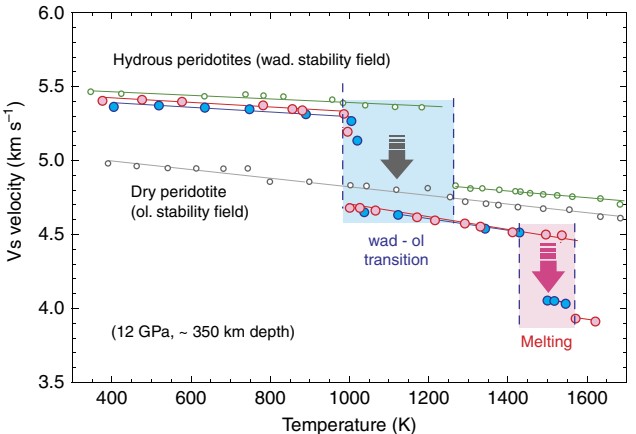

**Fig. 1** Secondary wave velocity as a function of temperature for hydrous peridotites. Hydrous peridotites synthesized in the wadsleyite stability field (14 GPa) and then re-equilibrated in the olivine stability field (12 GPa) demonstrate two distinct velocity drops: the first corresponding to the wadsleyite-to olivine transition at low temperature, and the second corresponding to the onset of melting at higher temperature. The hydrous peridotite with low-water content indicates velocity decrease corresponding to the wadsleyite-to-olivine phase transformation. The velocity variation of dry peridotite, synthesized at olivine stability field, does not indicate any abrupt variation in sound wave velocity and follows the trend for general decrease of bulk and shear moduli with temperature. The blue, red, and green circles represent hydrous wadsleyite with 3600, 3000, and 600 wt. p.p.m. of $H_2O$, synthesized at wadsleyite stability field. The gray circles represent dry peridotite synthesized at olivine stability field

## Results

**Sound wave velocity.** At a fixed pressure of 12 GPa, the sound velocity of peridotite before melting is characterized as expected by a gradual decrease of compressional and shear wave velocities with increasing temperature. For peridotite containing 600 p.p.m. wt. $H_2O$, we observe one abrupt drop of velocity corresponding to the wadsleyite-to-olivine phase transformation (Fig. 1). For peridotite samples with >3000 p.p.m. wt. $H_2O$, we observe two distinct velocity perturbations: a first velocity drop at 1000–1100 K and a second at 1450–1500 K (Fig. 1). For comparison, the dry peridotite sample (synthesized in the olivine stability field) does not show any abrupt variation in sound wave velocity (Fig. 1).

The first velocity perturbation observed upon increasing temperature for hydrous peridotite samples synthesized in the wadsleyite stability field is explained by the back-transformation of wadsleyite into olivine. The second velocity drop observed at around 1450 K exclusively in the most hydrous peridotites (>3000 wt. p.p.m. $H_2O$) can be attributed to the onset of melting. In the temperature range between the two velocity jumps, the released fluid is likely incorporated back into the remaining wadsleyite fraction.

**Chemical and micro-texture analysis.** Analysis of experimental run products after the velocity measurements had been completed confirmed the presence of melt in $H_2O$-bearing samples (Fig. 2), while no sign of melting was observed in the $H_2O$ undersaturated peridotite sample with 600 wt. p.p.m. $H_2O$. Chemical analyses of the melt show a significant enrichment in incompatible elements: Ca, Al, Na, K, together with increasing $H_2O$ and FeO contents with decreasing melt fractions (Supplementary Table 1).

The melt is distributed in a network of interconnected grain boundary tubes and melt-filled triple junctions within the host

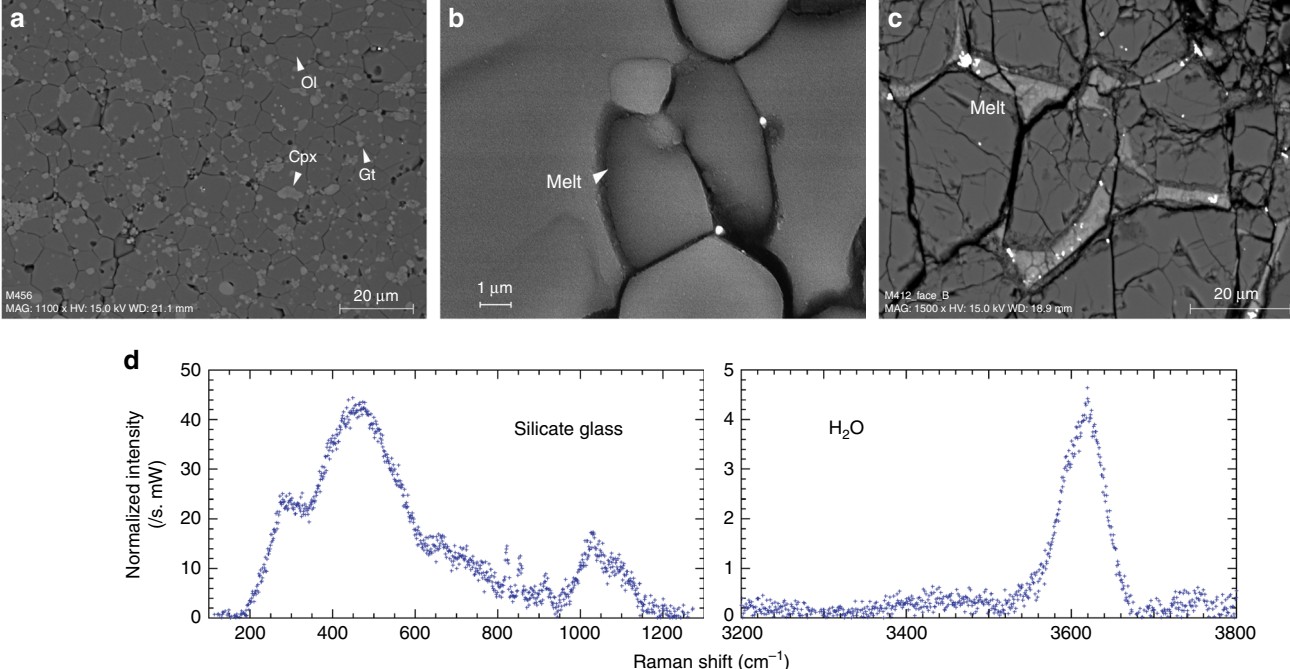

**Fig. 2** Sample characterization after high-pressure, high-temperature experiments. **a–c** FEG-SEM images showing mineral and melt distribution in recovered samples. **d** Micro Raman spectra of preserved hydrous silicate glass. Low-frequency bands (200–1200 cm$^{-1}$) correspond to vibrations of the alumino-silicate network (weak olivine peaks at 824 and 856 cm$^{-1}$ overlap the silicate glass spectrum); High-frequency region (3200–3800 cm$^{-1}$) corresponds to O-H stretching vibrations of OH groups and $H_2O$ molecules

mineral matrix (Fig. 2). Using the high-resolution SEM images, we determined the sample melt fractions, as well as the wetting angles of melt-solid interfaces for samples with different melt fractions. Dihedral angles are <10° for all measured samples and the median angle slightly increases with increasing melt fraction (Fig. 3).

The small dihedral angles observed in the present melts (Fig. 3) ensure complete grain boundary wetting and melt interconnectivity even for extremely low-melt volume fractions[34–36], which is crucial for propagation of seismic waves. The solid-melt dihedral angle is known to vary with pressure and with the composition of the melt phase[22,37]. The slight increase of dihedral angle with increasing melt fraction (Fig. 3) is a unique observation and can be explained by the decrease in $H_2O$ content in melt at higher degrees of melting. Due to the high wetting properties of the melt, we assume that slight changes in dihedral angle with melt fraction may not affect the seismic wave velocity in our samples.

## Discussion

Quantitative comparison between laboratory data and seismological signals requires experiments in which the molten phase is in textural equilibrium with the solid matrix. Due to time-limited laboratory experiments, transient conditions may affect the results of sound velocities. Therefore, texture analysis and interpretation are important. In a partially molten system loaded at given pressure and temperature, the melt network can evolve to minimize the energy of melt-solid interfaces. This equilibration process concerns the wetting angle $\theta$ at solid–solid-melt triple junctions, the area-to-volume ratio of melt pockets at grain corners and the melt permeability threshold[34,35]. Experimental studies suggest that textural maturation is a time-dependent process, which usually requires long annealing times (weeks or months)[35,38]. Still, the near-zero dihedral angle ($\theta < 10°$), the extensive wetting of crystal faces and the smoothly curved solid-melt interfaces observed in our samples are strong indications that the

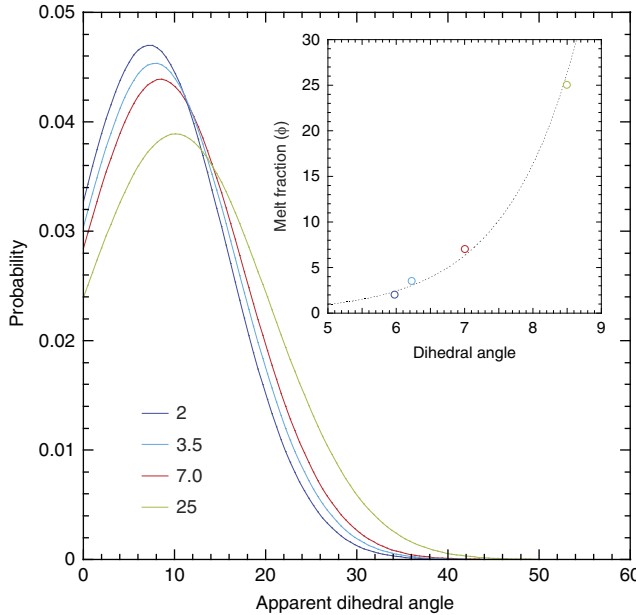

**Fig. 3** The variation of dihedral angle with the melt fraction. The probability curves indicate a slight increase in dihedral angle with increasing melt fraction. The majority of melt occurs as melt-filled grain boundaries. A total of more than 50 measurements were used for the fitting. (Inset) the change of dihedral angle with the melt fraction. The dihedral angle for the 0.7% melt fraction should be <5°

microstructure has reached an equilibrium[39–41] (Fig. 2). Further, after reaching the peak temperature, the sound velocity remains nearly constant, suggesting that the samples are well relaxed, enabling a safe comparison of our measurements with geophysical observations.

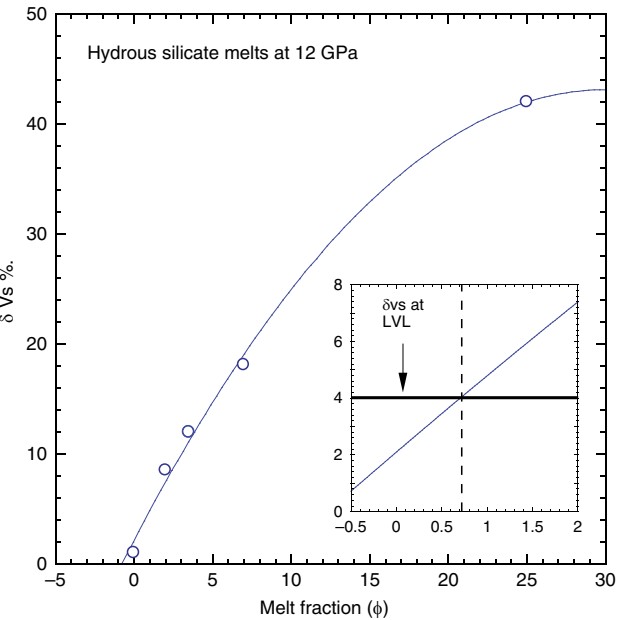

**Fig. 4** The % drop in secondary wave velocity as a function of sample melt fraction. The second order polynomial fitting indicates the best fitting line up to 25% melt fraction. The inset figure indicates the melt fraction required to explain the geophysically observed −4% δVs, yielding 0.7% of melt fraction for the LVL above the mantle transition zone

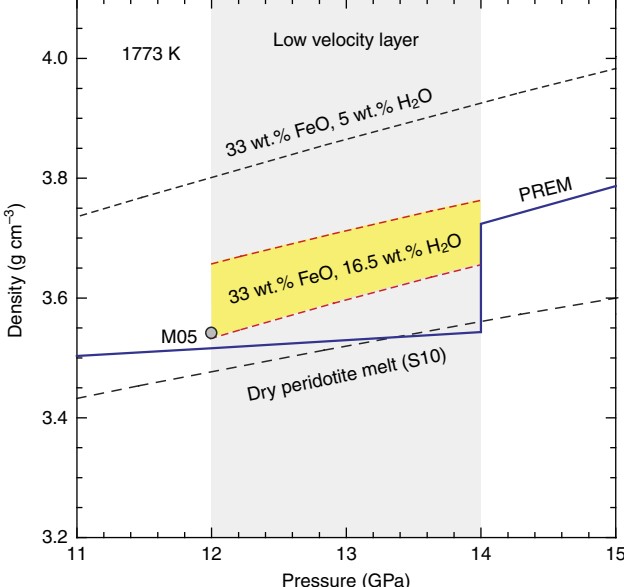

**Fig. 5** Density of hydrous silicate melts occurring at the base of the Earth's upper mantle. The density of hydrous melt is determined by the third-order Birch–Murnaghan equation of state (see Methods section for details). The yellow shaded area indicates the possible range of densities obtained for the 0.7% melt fraction with 33 wt.% FeO and 16.5 wt.% $H_2O$. The calculated density for melt with 33 wt.% FeO and 5 wt.% $H_2O$ in indicated in black broken line. The experimental data for hydrous rich melt[27] (M05) and dry peridotite melt[62] (S10) are shown for comparison. The mantle densities are taken from PREM[45] with corrections applied for the thermal expansion

We pointed out the change in melt chemistry with the degree of melting of our peridotite samples. Recent studies have shown that such compositional variations do not influence the propagation of seismic waves significantly[42,43]. Accordingly, based on the sound wave velocity of our melt-free and melt-bearing samples, we derive a relation between the % drop in shear wave velocity and the sample melt fraction. According to this model, the negative 4% shear wave velocity anomaly of the mantle can be explained by the presence of 0.7% volume fraction of hydrous silicate melt in the LVL (Fig. 4). Regional variations in the melt fraction, possibly induced by lateral temperature variations, could cause variable Vs velocity anomalies, reported elsewhere, in the LVL[2,6].

Our laboratory experiments demonstrate that partial melting can be triggered by upwelling of hydrous transition zone material. Still, the stability of a melt layer at the base of the upper mantle is critically linked to its density[23]. Based on the chemical composition of the hydrous silicate melts obtained in this study, we determined the melt density using the third-order Birch–Murnaghan equation of state, taking into account their concentrations of $H_2O$, FeO, $CO_2$, $Na_2O$, and $K_2O$[23,27,44] (See Methods section for details). Our calculations suggest that the presence of incompatible elements such as $CO_2$, $Na_2O$, and $K_2O$ have a negligible effect on melt density (<1% variation for the maximum possible abundances) compared to FeO and $H_2O$. The melt with 33 wt.% FeO and 16.5 wt.% $H_2O$, which corresponds to a melt fraction of 0.7 vol%, presents a density of 3.58–3.74 g cm$^{-3}$ in the pressure range of 12–14 GPa (~350–410 km depth) (Fig. 5). For comparison, the density of the ambient mantle varies from 3.51 to 3.72 g cm$^{-3}$[45] indicating that the melt is neutrally buoyant at those depths.

The stability of the melt layer at the base of the upper mantle is determined by the efficiency of melt generation and melt extraction processes. Hydration of MTZ mineral phases is key and the downwelling slabs can continuously provide a subsequent amount of water to the MTZ due to entrainment of hydrous

phases from the lithosphere[46]. The transition zone water filter (TZWF) model[1,23] details possible interactions between a melt layer and the surrounding mantle. For example, melt entrainment in downwelling slab components provides a plausible mechanism for recycling of $H_2O$ and incompatible elements back to the MTZ and the lower mantle. The lateral variations in temperature[47], water content and/or the rate of mantle upwelling could affect significantly the degree of partial melting in the LVL, and therefore the melt density. For example, relatively hotter regions should produce more melt with lower density than those subjected to smaller degrees of melting. Even subtle variations in melt density could result in upward or downward movement of melt, toward the Earth's surface or back to the MTZ, respectively. Such melt migration could be a controlling parameter of the LVL thickness.

In this study, we show that a melt fraction of 0.7 vol.% is compatible with the seismological observations in the LVL. With the working hypothesis of melt fraction in the LVL being similar to the degree of partial melting (i.e., the batch melting scenario), we estimate the water content in the source material. Our calculations show that 0.7 vol.% melt with 33 wt.% FeO requires 16.5 wt.% $H_2O$ in order to be gravitationally stable at 350–410 km depth. Knowing the $H_2O$ partitioning coefficient between the melt and the peridotite residue[48], we calculated the $H_2O$ contents in the MTZ to be $0.22 \pm 0.02$ wt.%. These water concentration values are in good agreement with previous estimations based on electrical conductivity[8,49]. However, variation of melt fraction within the LVL may indicate regional variations of water content in the MTZ. The water depleted[50] or water saturated models for the MTZ[51,52] cannot be corroborated by the findings of the current study. Particularly, the near-water saturated conditions implied by hydrous ringwoodite inclusions found in natural diamond[52] may not be representative of the whole MTZ but

instead may represent an episode of subduction-zone-related melting.

## Methods

**Sample preparation**. Starting materials with a composition similar to KLB-1 peridotite[53] were prepared using regent grade oxides, initially mixed in the absence of the required amount of Al(OH)$_3$ to yield the correct mineral composition. These powders were dried at 300 °C overnight to remove any adsorbed moisture. Na and K were added as NaCO$_3$ and K$_2$CO$_3$ and then decarbonated. The decarbonation of the oxide mixtures was carried out by slowly increasing the temperature to 1000 °C (1.6 °C min$^{-1}$) and keeping it at 1000 °C for about 10 h to ensure complete decarbonation. The decarbonated powder mixture was then cooled to 200 °C and stored in a high vacuum furnace at 120 °C prior to the hot pressing runs. The required amount of Al(OH)$_3$ was then mixed with the decarbonated powder mixture to obtain the desired amount of water in peridotite samples. The resulting powder mixtures were hot pressed to obtain solid sintered samples for seismic velocity measurements. The dry samples were synthesized at 5 GPa and 1473 K for 2 h in rhenium (Re) foil capsules. The low-water storage capacity in peridotite at this pressure[54] helps maintain relatively water-poor conditions in the nominally dry sample. Hydrous samples were synthesized at 14 GPa and 1373 K for 2 h in Gold–Palladium (Au–Pd) capsules. Given that the water storage capacity of wadsleyite is 4–5 times higher than olivine[54] and that the water storage capacity of clinopyroxene and garnet increases with pressure, the synthesis of hydrous samples at 14 GPa allows incorporation of more water into minerals phases. The wadsleyite-bearing sample with relatively low-water content was synthesized by first hot-pressing in the olivine stability field (2.5 GPa and 1473 K) to lower the initial water content in minerals, and then by hot-pressing the pre-sintered sample at 14 GPa and 1473 K in an Au–Pd capsule. Si-metal powder was also used to surround the capsules in order to reduce the influence of adsorbed moisture on the synthesis process. Water under-saturated conditions are required during the synthesis process, in order to avoid hydrous silicate melts along grain boundaries, which could otherwise interfere with the conductivity and sound wave velocity. Cylindrical core samples ~1.0 mm in length and ~1.2 mm in diameter were prepared from these pre-synthesized sample specimens. In situ measurements were performed on double-polished core samples of the starting material. The final length and diameter of the samples were measured with an accuracy of ±0.1 μm. The chemical composition and the water contents of the starting samples were analyzed using electron microprobe and micro-Raman, respectively.

**High-pressure and high-temperature experiments**. Simultaneous high-pressure and high-temperature experiments were performed using a 1500-ton Kawai-type multi-anvil apparatus at the Laboratoire Magmas et Volcans, Clermont-Ferrand, France. Both sample synthesis and the in situ measurements were performed using an octahedral pressure medium composed of MgO and Cr$_2$O$_3$ (5 wt.%) in a 14/8 multi-anvil configuration (octahedron edge length/anvil truncation edge length) For in situ measurements, the assembly was designed to accommodate the geometrical requirements for measurements of Vp, Vs, and EC within a single high-pressure cell. The pre-synthesized cylindrical sample was inserted into a single-crystal MgO sleeve. The sleeve also helps to insulate the sample electrically from the furnace. This furnace, composed of a 50 μm thick cylindrical Re foil, has apertures for the electrode and the thermocouple wires. A zirconia sleeve around the furnace was used as a thermal insulator.

We placed two electrodes, made of Re discs, at the top and bottom of the cylindrical sample. A tungsten-rhenium (W$_{95}$Re$_5$-W$_{74}$Re$_{26}$) thermocouple junction was placed at one end of the sample to monitor the temperature. The opposite end was connected to a single W$_{95}$Re$_5$ wire. We collected impedance spectra between the two W$_{95}$Re$_5$ wires. A dense Al$_2$O$_3$ buffer rod was placed between one of the tungsten carbide (WC) anvil truncations and the sample to enhance the propagation of elastic waves and to provide sufficient impedance contrast to reflect ultrasonic waves at the buffer rod-sample interface. Both ends of the anvil, the alumina buffer rod and the samples were mirror-polished using 0.25 μm diamond paste in order to enhance mechanical contacts. All ceramic parts of the cell assembly, including the pressure medium, were fired at 1373 K prior to their assemblage in order to remove any adsorbed moisture, and kept (as the samples) in vacuum furnaces (10$^{-2}$ Torr and 150 °C) before assembling the experiment. Oxygen fugacity of the sample was not controlled during in situ measurements, but was expected to remain below the Mo-MoO$_2$ buffer.

The upwelling of the hydrous mantle transition zone across the 410 km discontinuity was simulated by synthesizing hydrous samples at conditions corresponding to the wadsleyite stability field (14 GPa), and then reloading them to 12 GPa (conditions corresponding to ~350 km in depth) in order to observe their melting behavior by monitoring their Vp, Vs, and electrical conductivity responses. Thus, more water, exceeding the storage capacity of olivine, can be incorporated as a point defect dissolved in the structure of wadsleyite. This procedure also ensures that a hydrous melt does not form during the synthesis process. Upon re-equilibration in the olivine stability field, the wadsleyite transforms to olivine and the new mineral assemblage cannot store the original water content; consequently a free fluid is liberated that triggers melting. By changing the initial water content of the wadsleyite sample or by increasing the sample temperature, we observe seismic

velocity responses of five different samples with melt fractions of 0, 2, 3.5, 7, and 25%. For comparison, we have also measured seismic wave velocity of anhydrous peridotites at 12 GPa pre-synthesized, guaranteeing melt-free conditions in water under-saturated upper mantle.

**Sound wave velocity measurements**. Sound wave velocities of the samples were measured using the ultrasonic interferometry technique[55]. In this method, electrical signals of sine waves of 20–50 MHz (3–5 cycles) with $V_{peak-to-peak}$ of 1–5 V were generated by an arbitrary waveform generator (Tektronix AFG3101C); they were then converted to primary ($V_P$) and secondary ($V_S$) waves by a 10 ° Y-cut LiNbO$_3$ piezoelectric transducer attached to the mirror-polished truncated corner of a WC anvil. The resonant frequency of the transducer is 50 MHz for compressional waves (P-waves) and 30 MHz for shear waves (S-waves). Elastic waves propagated through the tungsten carbide (WC) anvil, alumina buffer rod (BR), and the sample, and were reflected back at the interfaces between the anvil-BR, the BR-sample, and the sample-electrode. The reflected elastic waves were converted back to electrical signals by the transducer and captured by a Tektronix DPO 5140 Digital Phosphor Oscilloscope at a rate of $5 \times 10^9$ sample per second. Signals at 20, 30, 40, and 50 MHz were recorded at each temperature step. The two-way travel time for the sound waves propagating through the sample can be determined by the time difference between the arrivals of the echoes from the BR-sample interface and the sample-electrode interface by the pulse-echo overlap method[55]. Sample lengths, prior to sample loading and after the melting experiments, were determined with a high precision digital height-gauge (accuracy of 0.1 μm) and using the BSE images obtained using Field Emission Gun scanning electron microscope (FEG-SEM), respectively. The change of sample length due to the thermal expansion at high temperature was corrected using the equations of state of olivine and wadsleyite, the principal mineral phases present in the samples.

**Chemical and micro-structural analyzes**. The chemical composition of pre-sintered samples and experimental run products after seismic velocity measurements were investigated using the Cameca SX100 electron probe micro analyzer at the Laboratoire Magmas et Volcans of Clermont-Ferrand. Energy-dispersive X-ray spectroscopy (EDS) chemical mapping was used to determine the mineral proportions. Micro-textures of the samples were observed with a Scanning Electron Microscope (SEM) JEOL Jeol JSM-5910 LV at the Laboratoire Magmas et Volcans of Clermont-Ferrand. For imaging, an accelerating voltage of 15 kV and working distance of 11.4 mm were used, and for chemical mapping an accelerating voltage of 10 kV and working distance of 19.3 mm were used. The fine melt micro-textures were observed with ZEISS supra 55VP field emission gun (FEG) SEM with an acceleration voltage of 15 kV and working distance of 9.7 mm (2MATech, Aubière, France).

Powder X-ray analysis using Philips PW 1830 (Cobalt wave-length) was carried out prior to the seismic velocity measurements to ensure the absence of additional hydrous phases, such as super hydrous phase-B in pre-sintered samples. No evidence for additional hydrous phases was found in the pre-synthesized hydrous peridotite samples.

**Water content estimations**. Raman spectroscopy was used to estimate the water contents of mineral and melt phases. Raman spectra were collected using an InVia confocal Raman micro spectrometer manufactured by Renishaw, equipped with a 532 nm diode laser (output power of ~200 mW), a Peltier-cooled CCD detector, a motorized XY stage and a Leica DM 2500 M optical microscope, housed at the Laboratoire Magmas et Volcans (Clermont-Ferrand). Scattered light was collected by a back-scattered geometry; the laser power on the sample was reduced to ~9 or 16.5 mW and the slit aperture was set to 65 μm (standard confocality setting) or 20 μm (high-confocality setting). A ×100 objective and 2400 l/mm grating were used for the analyses. These analytical conditions result in spatial resolution of ~1 μm and spectral resolution better than 1 cm$^{-1}$. Daily calibration of the spectrometer was performed based on a Si 520.5 ± 0.5 cm$^{-1}$ peak. The spectra were recorded using Wire 4.2 Software from ~100 to 1300 cm$^{-1}$ (alumino-silicate network domain) and from ~3000 to 3800 cm$^{-1}$ (water domain). Acquisition times were 60–240 s for the high-frequencies domain and 30–60 s for the low frequencies. Raman analysis was performed on mineral phases (wadsleyite, olivine, pyroxene, and garnet) and on the interstitial glass phase. Difficulties in Raman analysis of the glass phase were caused by the very small size of glass pockets (sometimes <1 μm), general instability of the water- and iron-rich glass under the laser beam, and overlapping of the surrounding olivine Raman peaks on the silicate glass bands. For determination of water content in glasses, we used both the external calibration procedure[56], which is based on a set of hydrous basaltic glass standards[57] and the absolute intensities of the water band area, and an internal calibration procedure, based on the correlation between the glass-water concentration and the relative areas of the water and silicate Raman bands. The two methods gave comparable results. Water contents of the standards were determined using both FTIR and SIMS techniques. Standards were analyzed at the same conditions as the samples. Analytical precision calculated based on repeated daily measurements of standard glasses is generally better than 6% relative.

**Experimental uncertainties**. Experimental measurements are subjected to uncertainties originating from the estimations of temperature, pressure, sample dimensions, data fitting errors and estimation of melt chemistry. The estimated experimental errors on the absolute values sound wave velocity are within 5% and it is <6% for chemical analysis.

**Calculations of silicate melt density at high pressure**. We estimated the density of silicate melts (compositions similar to the peridotite melts at 10–20 GPa pressure range) and found that the most important melt oxides that control peridotite density ($SiO_2$ content is roughly constant) at high pressures are FeO and $H_2O$, consistent with previous studies[27,44,58].

The first step of our calculations was to determine the density of dry peridotite and hydrous peridotite melt (5 wt.% $H_2O$) at high pressure, in order to estimate the effect of $H_2O$ on melt density in the 10–20 GPa pressure range. We used the third-order Birch–Murnaghan equation of state (EOS):

$$P = \frac{3}{2} K_T \left( \left( \frac{\rho}{\rho_0} \right)^{\frac{7}{3}} - \left( \frac{\rho}{\rho_0} \right)^{\frac{5}{3}} \right) \times \left( 1 - \frac{3}{4}(4 - K') \times \left\{ \left( \frac{\rho}{\rho_0} \right)^{\frac{2}{3}} - 1 \right\} \right), \quad (1)$$

where $\rho$ is the high-pressure density, $\rho_0$ is the zero-pressure density, $K_T$ is the isothermal bulk modulus and $K'$ is its pressure derivative. The temperature effect on $K_T$ is expressed with the following equation:

$$K_T = K_{T_0} + \left( \frac{\partial K_T}{\partial T} \right)_P (T - T_0), \quad (2)$$

with the following parameters[44] used for hydrous peridotite (5 wt.% $H_2O$) at 1773 K: $\rho_0 = 2.40$ g/cm$^3$, $K_T = 8.8 \pm 1.9$ GPa, $K' = 9.9 \pm 3.6$, and $\left( \frac{\partial K_T}{\partial T} \right)_P = -0.0022 \pm 0.0015$(GPa/K) and the following parameters for dry peridotite at 2100 K: $\rho_0 = 2.72$ g/cm$^3$, $K_T = 24.0 \pm 1.3$ GPa, $K' = 7.3 \pm 0.8$, and $\left( \frac{\partial K_T}{\partial T} \right)_P = -0.0027 \pm 0.0017$(GPa/K).

We found that $H_2O$ decreases the density of peridotite melt in the 10–20 GPa pressure range at 1770 K by about 0.02 g cm$^{-3}$ for every 1 wt.% $H_2O$ This result is in good agreement with the previous studies in the pressure range of 10–16 GPa[27]. However, for lower pressures (1–5 GPa pressure range), water decreases the density of silicate melt by about 0.04 g cm$^{-3}$ for every 1 wt.% $H_2O$[59]. This is consistent with the fact that water is more compressible than the major oxides of silicate melts at upper mantle conditions ($SiO_2$, $Al_2O_3$, FeO, MgO, and CaO) (see Matsukage et al.[27] for more details).

The computed partial molar volume of $H_2O$, $H_2O = 7.6$ cm$^3$ mol$^{-1}$ at 15 GPa, is in good agreement with the ~$(8 \pm 2)$ cm$^3$ mol$^{-1}$ of Matsukage et al.[27].

The effect of iron was examined by using the data of Matsukage et al.[27] at pressure and temperature conditions for a depth of 410 km. Supplementary Fig. 1 reports the variation of the isothermal bulk modulus ($K_T$) as a function of the FeO content in silicate melt (mol.%). For the value of $K'$, although its value must be >4 for silicate melts at high pressure[60], we assumed a value of 4 in our calculations (mainly due to the limited experimental data on silicate melts at high pressure).

To estimate the effects of $Na_2O$, $K_2O$, and $CO_2$ on the density of silicate melt at high pressure, we used the reported partial molar volumes[23] in conjunction with the volume of hydrous peridotite melt (5 wt.%). In the 10–20 GPa pressure range, adding a total of up to 1 wt.% of these oxides does not drastically change the density of the silicate melt. For instance, adding ~2000 p.p.m. wt. $CO_2$ (maximum possible value for our composition, based on Raman spectra), ~2000 p.p.m. wt. $Na_2O$ and ~8000 p.p.m. wt. $K_2O$ to a hydrous peridotite melt (with 5 wt.% $H_2O$) will lower the density from about 3.5 to 3.47 g cm$^{-3}$ at 16 GPa. So, in the following calculations we take into account only the effects of $H_2O$ and FeO on the density of silicate melts at high pressure, and ignore the effects of the other oxides.

To determine the density of our melt (that contains 20.2 mol.% FeO) at high pressure we used the following steps:

We determined the density of our Fe-rich hydrous melt (~14 mol.% $H_2O$) at high pressure using the Birch–Murnaghan EOS with $K' = 4$ and $K_T = 17.462 - 0.12154 \times$ FeO (mol.%)

$$P = \frac{3}{2} K_T \left( \left( \frac{\rho}{\rho_0} \right)^{\frac{7}{3}} - \left( \frac{\rho}{\rho_0} \right)^{\frac{5}{3}} \right). \quad (3)$$

$\rho_0$ of our composition was determined using the Ochs and Lange[61] model at 1 bar and 1773 K.

Then we estimated the effect of $H_2O$ (up to 40 mol.%) on the density of the Fe-rich hydrous melt by using the partial molar volume of water:

$$V_{FeO \text{ and } H_2O-rich \text{ melt}} = (1 - \delta X_{H_2O}) \times V_{FeO-rich \text{ melt}} + \delta X_{H_2O} \times {}_{H_2O}, \quad (4)$$

where $\delta X_{H_2O}$ is the difference in molar fraction of water between the FeO-$H_2O$-rich melt and the FeO-rich melt.

Another way to estimate the effect of water on the density of silicate melts at high pressure is to assume (as noted above) that $H_2O$ decreases the density of silicate melts in the 10–20 GPa pressure range at 1770 K by about 0.02 g cm$^{-3}$ for every 1 wt.% $H_2O$:

$$\rho_{FeO \text{ and } H_2O-rich \text{ melt}} = \rho_{FeO-rich \text{ melt}} - 0.02 \times \delta X_{H_2O}, \quad (5)$$

where the $\delta X_{H_2O}$ is the difference in wt.% of water between the FeO–$H_2O$-rich melt and the FeO-rich melt.

**Data availability**. All data generated or analyzed during this study are included in this published article and its Supplementary Information files. The raw data in digital format can be obtained from the corresponding author on reasonable request.

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

## Acknowledgements

We thank S-I Karato and an anonymous reviewer for their constructive reviews that benefited the manuscript. G.M. acknowledges funding from the French PNP program (INSU-CNRS). NBC is supported by ANR-11-JS56-01501 and D.A. is supported by ANR-13-BS06-0008. This research was financed by the French Government Laboratory of Excellence initiative n°ANR-10-LABX-0006, the Région Auvergne and the European Regional Development Fund. This is ClerVolc contribution number 274.

## Author contributions

G.M. conceived the idea, designed the experiments and wrote the paper. D.F. and G.M. performed the experiments. D.F. and G.M. analyzed the data. F.S. and N.B.C. helped with the Raman and FTIR analyzes. J.C. helped with the analysis of sound velocity data. M.A. B. performed the density calculations. D.A. contributed to the discussion. All authors participated in the discussion and agreed on the content.

## Additional information

**Competing interests:** The authors declare no competing financial interests.

