## [Peer Review File · Nature Communications]

Reviewers' comments:

Reviewer #1 (Remarks to the Author):

The manuscript presents laboratory measurements of the geophysical and geochemical properties of hydrated and dry peridotite at conditions of the deep upper mantle where dehydration melting may occur in upwelling mantle. The measurements are impressive and could have an important impact on interdisciplinary understanding of mantle composition and dynamics. It is valuable to have experimental results at multiple levels of hydration and low melt fractions along with constraints on shear velocity and density. However, the implications of the new results are weakly developed in this very brief text and there are quite a minor typos and awkward sentences. So, the body of results is much more impressive than the written presentation. The figure quality is good. I think the results could certainly merit publication in a very high impact journal, but I would encourage the authors to expand the manuscript beyond its present cursory state. I think this could be achieved in framework of 'minor revisions'. Some more detailed comments are given below.

25. these studies were depicting potential based on 1 isolated sample and a suite of lab experiment, so it would be fairer to say something like ... significantly or about an order of magnitude lower than the storage capacity.

63-70. It would be helpful to mention the pressure conditions in the text here. It would help some readers understand why there is not an olivine-to-wadsleyite transition for the dry case.

73-79. It would be helpful to mention how many different melt fractions were observed, it appears to be 5 based on the figures. How did the melt geometry (estimated dihedral angle) change among these samples?

98. 0.7 '%' ?

109-113. Is this is all based on assumption of 4% Vs reduction? If so, the uncertainties are much too small as the low-Vs layer does not exist everywhere and the Vs reduction is variable. It would be hard/impossible to precisely quantify these effects, but they should be noted because it's not as though this study constrains the actual H₂O content within +/- 0.04 wt. %. I agree that the results are a powerful indication that actual hydration levels in the transition zone are on the order of tenths of a percent.

Could the authors give any further insights regarding how plumes or slabs would interact with the melt layer? Or how the melt geometry changed with the degree of hydration or melt fraction and the resulting implications for visibility of the melt layer? Or how the incompatible element depletion due to melting compares with estimates of the upper mantle sampled by mid-ocean ridges? Or how the dramatic variation in the olivine-wadsleyite transition compares to observational evidence? ...

There are many topics raised in the introduction (which is also unfortunately brief) that are not even touched upon in the discussion. There's plenty of room to do more with this manuscript in the Nature Communications format.

Reviewer #2 (Remarks to the Author):

General comments

This paper provides important experimental observations that are highly relevant to the interpretation of the seismologically inferred low velocity regions above the 410-km.

The experiments have been done more-or-less properly, and the results are well documented in most cases (I will discuss a few important technical issues however). And the most important results are the measurements of shear wave velocity as a function of melt fraction.

One major problem, however, is the ignorance of highly relevant papers including [Karato et al., 2006], [Yoshino et al., 2007], [Karato, 2011], [Jing and Karato, 2009], [Jing and Karato, 2011] and [Jing and Karato, 2012] where the problems studied in this paper were studied in detail. This is truly surprising. In many cases, the results shown in this paper confirm the results from these earlier studies. However, ignoring these highly relevant papers is not accepted in the scientific community.

The authors claim that there were no detailed studies on the densities of hydrous peridotitic melts. This is incorrect. [Matsukage et al., 2005], [Jing and Karato, 2011] and [Jing and Karato, 2012] conducted detailed studies on peridotitic melts (including water) in order to address the question of the nature of melt at above the 410-km (see also Matsukage et al., 2005). [Yoshino et al., 2007] investigated the wetting behavior of (hydrous) melt down to the deep upper mantle conditions. [Karato, 2011] inferred water content of the transition zone to conclude that on average it contains ~ 0.1 % of melt with large regional variation, a conclusion in good agreement with the results of this present work (~ 0.2 % melt) within the uncertainties (a factor of 2 or so: velocity reduction is not exactly 4%. Its value varies among different regions, but also it depends on the method that seismologists use (e.g., the frequency of seismic waves etc.)). These previous papers must be cited and the authors must discuss how their own works compare with these previous studies. Such a necessary step is not made in this paper.

Another fundamental issue is that the authors do not distinguish the degree of melting from the melt fraction. These two are different, and mixing up these two cannot be justified. To understand this important question, one may consider a well-known case of melting in the mid-ocean ridge. The degree of melting associated with the formation of the oceanic crust is $\sim 10\%$. But the melt fraction below a major part of a mid-ocean ridge is ~ 0.1 % (estimated from the geochemical observations [Spiegelman and Elliott, 1993] and also from a simple model calculation of melt transport [Karato, 2014]). The degree of melting is controlled by thermodynamics, but the melt fraction is controlled by the processes of melt transport. These two agree only when melt does not move.

Specific comments

I will list several specific issues that need to be clarified or corrected.

(1) On the melt density

(1-1) The influence of chemical composition on melt density is well known and studied by many other scientists. A particularly important is the influence of "other" volatile components. All volatile components (H, C, Na, K etc.) affect the solidus in a similar way when the influence is parameterized in terms of the atomic fraction (effects through a change in configurational entropy). However, the influence of these species on melt density differ substantially simply because these elements (for a given atomic fraction) have different mass. Such an analysis was made by us (e.g., [Karato et al., 2006]), and reference needs to be made, and they should provide some comments.

(1-2) The validity of calculations of the melt density in this paper is unclear. First, they did not measure the melt density. They used previous reports on the density of dry and water-saturated (wet) melts by [Sakamaki et al., 2009] and calculated the melt density with intermediate values of water content using the Birch-Murnaghan equation of state, I think (it is unclear what they did). As far as what they did is just interpolating the data, the method to interpolate would not matter so much. For instance, when one uses the Birch-Murnaghan equation of state, one needs ρ_0 , K_0 and K'_0 for an intermediate values of water content and various FeO content. How could one make such a guess? If one takes a simple linear interpolation in these parameters, why not using a linear interpolation of density at high P-T for two end-member compositions? One might use a little more physical model such as a model by [Jing and Karato, 2012]. Clarification is needed as to how they calculated the density for various compositions at various P-T conditions.

(2) On the dihedral angle

The dihedral angle is an important parameter to control the velocity of seismic waves in a partially molten material. However, a key step in determining the dihedral angle is to make sure that a sample reaches textural equilibrium (at least approximately). In some cases, it takes long time to reach equilibrium, and if a sample texture (melt geometry) is not equilibrium, then the applicability of such results is dubious. This issue has been recognized at least for 35 years from the beginning of such a study by Harve Waff (in late 1970s). For instance, [Waff and Blau, 1982] annealed their samples to 720 hours, and showed the evolution of apparent dihedral angle up to ~200 hours. Obviously the time needed for near equilibrium geometry depends strongly on grain-size, but it is essential to confirm near equilibrium geometry. The authors should show evidence for near equilibrium geometry of melt.

(3) On the velocity measurements and their interpretation

They measured seismic wave velocities as a function of melt fraction. I find two major problems in this part (in addition to the issue of textural equilibrium discussed above).

(3-1) The first problem is related to the important distinction between the degree of melting and the melt fraction (the degree of melting and the melt fraction are different in the open system, and it is the melt fraction and not the degree of melting that affects seismic wave velocity). When one wants to show the influence of melt fraction on seismic wave velocity, it is better to prepare such a diagram for fixed (but different) melt composition. In contrast, if one conducts an experimental study in a closed system (most of lab studies are for a closed system) where melt fraction agrees with the degree of melting, then as melt fraction changes the melt composition also changes. Consequently, the application of such results will be complicated. There is no discussion on this issue.

(3-2) The second is the interpretation of their results in terms of a model. The influence of partial melting on seismic wave velocities has been studied by many, mostly theoretically (e.g., [O'Connell and Budianski, 1974; 1977; Stocker and Gordon, 1975; Takei, 2002]), but [Sato et al., 1989] published a paper on the experimental results at low P (0.5-1.0 GPa). These theoretical papers predict a relation between velocity reduction and melt fraction that depends on the wetting behavior (dihedral angle). ~4 % velocity reduction for ~0.7% melt would correspond to some dihedral angle. How does this result compare with those by Sato et al. (1989)? What dihedral angle do their results correspond to? Does it agree with [Yoshino et al., 2007]'s results? With this dihedral angle, would melt stay there or should it move down or up (what about compaction)? These discussions are important but I do not see any of them.

(4) On the influence of temperature on the melt density

On page 4 (lines 103-105), the authors present a discussion on the influence of temperature on melt density. They argue that at relatively hot mantle, relatively low melt density because the degree of melting is high and less FeO. But how about the influence of water content? At high temperature, the water content in the melt will be smaller making the melt density to be higher. One needs to compare the influence of both FeO and H₂O.

(5) On the conversion of the melt fraction to the water content in the transition zone

The authors use their results on the necessary melt fraction to explain the reduction of shear wave velocity to infer the water content in the transition zone. There is an important jump in the logic here. The magnitude of velocity reduction is determined by the melt fraction. The water content in the transition zone will determine the degree of partial melting. This is not the same as the fraction of melt. These two agree only when melt stays the place where it is formed (batch melting). A discussion is necessary to support their conclusion on the water content in the transition zone from the inferred melt fraction in a layer above the 410-km.

In addition, there are many grammatical errors. I corrected some, but an editorial office must take a look at this issue.

I believe that this paper contains a very useful data set, but the interpretations contain several

fundamental issues. I hope that the authors can address the comments/questions listed above.

Shun-ichiro Karato

References

- Jing, Z., and S. Karato (2009), The density of volatile bearing melts in the Earth's deep mantle: The role of chemical composition, *Chemical Geology*, 262, 100-107.
- Jing, Z., and S. Karato (2011), A new approach to the equation of state of silicate melts: An application of the theory of hard sphere mixtures, *Geochimica et Cosmochimica Acta*, 75, 6780-6802.
- Jing, Z., and S. Karato (2012), Effect of H₂O on the density of silicate melts at high pressures: Static experiments and the application of a modified hard-sphere model of equation of state, *Geochimica et Cosmochimica Acta*, 85, 357-372.
- Karato, S. (2011), Water distribution across the mantle transition zone and its implications for global material circulation, *Earth and Planetary Science Letters*, 301, 413-423.
- Karato, S. (2014), Does partial melting explain geophysical anomalies?, *Physics of the Earth and Planetary Interiors*, 228, 300-306.
- Karato, S., D. Bercovici, G. Leahy, G. Richard, and Z. Jing (2006), Transition zone water filter model for global material circulation: Where do we stand?, in *Earth's Deep Water Cycle*, edited by S. D. Jacobsen and S. van der Lee, pp. 289-313, American Geophysical Union, Washington DC.
- Matsukage, K. N., Z. Jing, and S. Karato (2005), Density of hydrous silicate melt at the conditions of the Earth's deep upper mantle, *Nature*, 438, 488-491.
- O'Connell, R. J., and B. Budianski (1974), Seismic velocities in dry and saturated cracked solids, *Journal of Geophysical Research*, 79, 5412-5426.
- O'Connell, R. J., and B. Budianski (1977), Viscoelastic properties of fluid-saturated cracked solids, *Journal of Geophysical Research*, 82, 5719-5735.
- Sakamaki, T., E. Ohtani, S. Urakawa, A. Suzuki, and Y. Katayama (2009), Measurements of hydrous peridotite magma density at high pressure using the X-ray absorption method, *Earth and Planetary Science Letters*, 287, 293-297.
- Sato, H., I. S. Sacks, and T. Murase (1989), The use of laboratory velocity data for estimating temperature and partial melt fraction in the low-velocity zone: Comparison with heat flow and electrical conductivity studies, *Journal of Geophysical Research*, 94, 5689-5704.
- Spiegelman, M., and T. Elliott (1993), Consequences of melt transport for uranium series disequilibrium in young lavas, *Earth and Planetary Science Letters*, 118, 1-20.
- Stocker, R. L., and R. B. Gordon (1975), Velocity and internal friction in partial melts, *Journal of Geophysical Research*, 80, 4828-4836.
- Takei, Y. (2002), Effect of pore geometry on V_p/V_s: From equilibrium geometry to crack, *Journal of Geophysical Research*, 107, 10.1029/2001JB000522.
- Waff, H. S., and J. R. Blau (1982), Experimental determination of near equilibrium textures in partially molten silicates at high pressures, in *High-Pressure Research in Geophysics*, edited by S. Akimoto and M. H. Manghnani, pp. 229-236, Center for Academic Publication, Tokyo.
- Yoshino, T., Y. Nishihara, and S. Karato (2007), Complete wetting of olivine grain-boundaries by a hydrous melt near the mantle transition zone Earth and Planetary Science Letters, 256, 466-472.

Response to reviewers' comments

We thank both reviewers for their in-depth reviews, which substantially improved the quality of the manuscript. The point-by-point responses to reviewer's comments follow in blue font.

Reviewers' comments:

Reviewer #1 (Remarks to the Author):

The manuscript presents laboratory measurements of the geophysical and geochemical properties of hydrated and dry peridotite at conditions of the deep upper mantle where dehydration melting may occur in upwelling mantle. The measurements are impressive and could have an important impact on interdisciplinary understanding of mantle composition and dynamics. It is valuable to have experimental results at multiple levels of hydration and low melt fractions along with constraints on shear velocity and density. However, the implications of the new results are weakly developed in this very brief text and there are quite a minor typos and awkward sentences. So, the body of results is much more impressive than the written presentation. The figure quality is good. I think the results could certainly merit publication in a very high impact journal, but I would encourage the authors to expand the manuscript beyond its present cursory state. I think this could be achieved in framework of 'minor revisions'. Some more detailed comments are given below.

1. Thank you for your constructive views. We have made significant improvements to both the introduction and the discussion. Details of these modifications / additions are summarized in response #7.

25. these studies were depicting potential based on 1 isolated sample and a suite of lab experiment, so it would be fairer to say something like ... significantly or about an order of magnitude lower than the storage capacity.

2. We have slightly modified the abstract text to accommodate reviewer's comment (L 22-23)

63-70 It would be helpful to mention the pressure conditions in the text here. It would help some readers understand why there is not an olivine-to-wadsleyite transition for the dry case.

3. The "dry peridotite" run was conducted at 12 GPa. But the sample was pre-synthesized at 5 GPa, in the stability field of olivine, where the water solubility in olivine is lower than at 12 GPa. This particular experiment serves as a benchmark for melt-free peridotitic upper mantle. Since the synthesis and both velocity measurements were done in the olivine stability field, the wadsleyite-to-olivine transition is not expected to occur. We have now modified the text to clarify this point. (L 79-80 & L 192-195)

73-79. It would be helpful to mention how many different melt fractions were observed, it appears to be 5 based on the figures. How did the melt geometry (estimated dihedral angle) change among these samples?

4. Total of 5 samples, 4 with melt and 1 melt-free, were used in this study to derive the relation between % Vs drop vs. the melt fraction. The different melt fractions were obtained by changing the initial water content of the starting materials or changing the sample temperature. This crucial information is now incorporated into the revised manuscript. (L 247-249)

The melt-sold dihedral angles are less than 10° for all melt fractions. However, the dihedral angle decreases with decreasing melt fraction. This can be explained by the effect of water on wetting properties of melt, as low melt fractions contain higher amount of water. We have included a paragraph (L 95-108) and a new figure (Figure 3) discussing this issue.

98. 0.7 '%' ?

5. Corrected in the revised manuscript.

109-113. Is this is all based on assumption of 4% Vs reduction? If so, the uncertainties are much too small as the low-Vs layer does not exist everywhere and the Vs reduction is variable. It would be hard/impossible to precisely quantify these effects, but they should be noted because it's not as though this study constrains the actual H2O content within +/- 0.04 wt. %. I agree that the results are a powerful indication that actual hydration levels in the transition zone are on the order of tenths of a percent.

6. In this study we have used the global average of 4% s-wave velocity drop observed by Tazuin et al. 2010 (nature comm. 3,718-721, 2010). Previous studies (Vinnik and Ferra, 2007, Revenaugh and Sipkin 1994) reported up to 7% Vs velocity drop in some regions. But such values were not corroborated by Tazuin et al. We believe that the seismic conversion method used by Tazuin et al. 2010 is superior for detecting and quantifying the seismic velocity anomalies compared to the previous studies. We agree that there could still be significant uncertainties associated with the estimations of velocity reductions.

Experimental measurements are subject to uncertainties originating from the estimates of temperature, pressure, sample dimensions, and data fitting errors and uncertainties related to water content estimations, which can be mounted up to 10% of the values. We have included this information in the revised manuscript. (L 322-326)

Could the authors give any further insights regarding how plumes or slabs would interact with the melt layer? Or how the melt geometry changed with the degree of hydration or melt fraction and the resulting implications for visibility of the melt layer? Or how the incompatible element depletion due to melting compares with estimates of the upper mantle sampled by mid-ocean ridges? Or how the dramatic variation in the olivine-wadsleyite transition compares to observational evidence? ...

There are many topics raised in the introduction (which is also unfortunately brief) that are

not even touched upon in the discussion. There's plenty of room to do more with this manuscript in the Nature Communications format.

7. We agree with the reviewer's comment about the lack of discussion of geophysical implications. In the revised manuscript, we have extended our discussion to several key points.

1. We have improved the discussion about wetting properties of hydrous silicate melts. The analytical results are now shown in a new figure in the main text (Figure 3). (L 95-108)
2. The geophysical and geochemical implications of a partially molten layer at the base of the upper mantle have been discussed extensively in previous papers (Bercovici and Karato 2003, Karato et al. 2006). While we cannot directly apply our experimental data to explain how this melt layer interacts with the surrounding mantle, we provide a brief discussion based on the transition zone water filter model proposed by Bercovici and Karato 2003 and Karato et al. 2006. (L 149-163)
3. In our experiments we observe the back transformation and the melting events in two separate temperature regimes (Figure 1). The velocity reduction associated with the back transformation of wadsleyite to olivine appears at low temperature and does not interfere with the melting event.

Reviewer #2 (Remarks to the Author):

General comments

This paper provides important experimental observations that are highly relevant to the interpretation of the seismologically inferred low velocity regions above the 410-km. The experiments have been done more-or-less properly, and the results are well documented in most cases (I will discuss a few important technical issues however). And the most important results are the measurements of shear wave velocity as a function of melt fraction.

One major problem, however, is the ignorance of highly relevant papers including [Karato et al., 2006], [Yoshino et al., 2007], [Karato, 2011], [Jing and Karato, 2009], [Jing and Karato, 2011] and [Jing and Karato, 2012] where the problems studied in this paper were studied in detail. This is truly surprising. In many cases, the results shown in this paper confirm the results from these earlier studies. However, ignoring these highly relevant papers is not accepted in the scientific community.

1. Thank you for your constructive views on our study. We realize that some prominent and highly relevant studies had not been cited or discussed in the previous version. We have corrected this issue in the revised manuscript. More than 30 relevant studies, including those you referenced, have been added to the revised manuscript.

The authors claim that there were no detailed studies on the densities of hydrous peridotitic melts. This is incorrect. [Matsukage et al., 2005], [Jing and Karato, 2011] and [Jing and Karato, 2012] conducted detailed studies on peridotitic melts (including water) in order to address the question of the nature of melt at above the 410-km (see also Matsukage et al., 2005). [Yoshino et al., 2007] investigated the wetting behavior of (hydrous) melt down to the deep upper mantle conditions.

2. We acknowledge that there have been many investigations into the density of melt at the 410-km discontinuity. What we implied in our previous version was that the melt compositions used in those studies were not determined from actual peridotite melting experiment performed at the conditions of the LVL. For example the FeO and H₂O contents of these studies were fixed and the densities were determined on somewhat arbitrary grounds. Since FeO and volatile components influence the melt density, we believe that a realistic melt composition is essential for accurate estimation. For example the FeO content that we obtain for our melt at 14 GPa is significantly higher than the FeO contents assumed in both Matsukage et al. 2005 and Sakamaki et al. 2006. We have slightly modified the manuscript text to clarify these points. (L 52-59)

In our previous version, we already used Matsukage et al. 2005 data to estimate the effect of FeO on the density of silicate melts at high pressure, (in the methods section of the previous version- the effect of FeO on the density of silicate melts at high pressures, line 295). Based on your comments (1-2), we have improved our density estimations using the systematics presented by Matsukage et al. 2005, Sakamaki et al. 2009 (for FeO and H₂O) and Karato et al. 2006 (for minor volatile components). (L136-148 in the main text and the revised methods section)

3. We have expanded our discussion on the wetting behavior of silicate melts and have included several absent studies, including that of Yoshino et al 2007. In addition, we have also included a new figure (Fig.3) in the revised manuscript showing the variation of dihedral angle with the melt fraction. (L 95-108)

[Karato, 2011] inferred water content of the transition zone to conclude that on average it contains ~0.1 % of melt with large regional variation, a conclusion in good agreement with the results of this present work (~0.2 % melt) within the uncertainties (a factor of 2 or so: velocity reduction is not exactly 4%. Its value varies among different regions, but also it depends on the method that seismologists use (e.g., the frequency of seismic waves etc.)). These previous papers must be cited and the authors must discuss how their own works compare with these previous studies. Such a necessary step is not made in this paper.

4. Thank you for pointing out this mistake. We have included the findings of Karato 2011 study in our revised manuscript, and modified the discussion accordingly. (L 173-174)

Another fundamental issue is that the authors do not distinguish the degree of melting from the melt fraction. These two are different, and mixing up these two cannot be justified. To understand this important question, one may consider a well-known case of melting in the mid-ocean ridge. The degree of melting associated with the formation of the oceanic crust is ~10%. But the melt fraction below a major part of a mid-ocean ridge is ~0.1 % (estimated from the geochemical observations [Spiegelman and Elliott, 1993] and also from a simple model calculation of melt transport [Karato, 2014]). The degree of melting is controlled by thermodynamics, but the melt fraction is controlled by the processes of melt transport. These two agree only when melt does not move.

5. We agree with the reviewer. This key issue was not addressed in our previous manuscript. The first point concerns our experimental data. The dihedral angles in our melts are extremely low and thus the melt should be fully interconnected even for very low melt fractions (as low as 0.01 %). The onset of melting and the melt interconnection should occur at the same time. Once we observed melting, we kept the sample at constant temperature, for at least one hour, so that melting would not progress further. There could still be subtle variations due to uncertainties in temperature estimations; however in our experimental setup the average degree of melting and the melt fraction are similar. The degree of melting controls the chemical fractionations, so that we can estimate the chemical variations in the melt and the residual solid, related to the hydrous peridotite source.

The second point is the application of laboratory data to mantle conditions.

1. For the seismic anomaly –Application of lab data to LVL should be straight forward as:
 - a. It has been reported that a small change in melt chemistry may not alter the seismic velocity in peridotite (e.g. Hier-Majumder et al. 2014, Afonso and Schutt 2012), so that small compositional variation at low degree melting may not matter. (L 126-134)
 - b. The wetting properties of melt remain similar for experimental and natural cases for the possible fractions in LVL (< 2%) because both deal with a hydrous peridotite system. (L 101-108)
2. Application to MTZ water contents- The chemical fractionation at melting is related to the degree of melting. In laboratory samples, estimations are straightforward; however, application to the mantle requires that the produced melt does not move out of its matrix/percolate following the batch melting. (L 167-169) This point can be fully corroborated with our experimental data. Nonetheless, we can discuss possible scenarios based on geophysical observations. For instance, both seismic and electrical profiles suggest that the LVL is restricted to the base of the upper mantle at the depth range 350-410 km. It indicates that pervasive upward percolation is not taking place. Also, the Poisson's ratio of the peridotite+melt system is small (0.31), indicating melt squirt due to compaction is not likely. The melt entrainment mechanism proposed by

the TZWF model may provide the best possible mechanism for the recycling of the melt layer back to MTZ.

Specific comments

I will list several specific issues that need to be clarified or corrected.

(1) On the melt density

(1-1) The influence of chemical composition on melt density is well known and studied by many other scientists. A particularly important is the influence of “other” volatile components. All volatile components (H, C, Na, K etc.) affect the solidus in a similar way when the influence is parameterized in terms of the atomic fraction (effects through a change in configurational entropy). However, the influence of these species on melt density differ substantially simply because these elements (for a given atomic fraction) have different mass. Such an analysis was made by us (e.g., [Karato et al., 2006]), and reference needs to be made, and they should provide some comments.

6. This is an important point. In our previous density calculations, we did not include the effect of C, Na and K oxides. This is simply because our calculations, based on both the measured melt chemistry and the volatile abundances in melt (Karato et al. 2006), indicated that the effect of such volatile components is significantly small on melt density (maximum 1 %).

However, we agree that this issue cannot be neglected in the discussion. The main text in the revised manuscript contains a discussion on the effect of these volatile components. Further, we have also updated the methods section with a detailed description of density estimations for Na, C and K. (L 139-145)

(1-2)The validity of calculations of the melt density in this paper is unclear. First, they did not measure the melt density. They used previous reports on the density of dry and water-saturated (wet) melts by [Sakamaki et al., 2009] and calculated the melt density with intermediate values of water content using the Birch-Murnaghan equation of state, I think (it is unclear what they did). As far as what they did is just interpolating the data, the method to interpolate would not matter so much. For instance, when one uses the Birch-Murnaghan equation of state, one needs rho-zero, K-zero and K'-zero for an intermediate values of water content and various FeO content. How could one make such a guess? If one takes a simple linear interpolation in these parameters, why not using a linear interpolation of density at high P-T for two end-member compositions? One might use a little more physical model such as a model by [Jing and Karato, 2012]. Clarification is needed as to how they calculated the density for various compositions at various P-T conditions.

7. We have carefully revised our density estimations. The discussion is presented in the methods section in the revised manuscript. With our calculations we were able to reproduce all existing experimental data for hydrous silicate melts in the pressure range of 10-16 GPa.

Our calculations are in good agreement with the estimations based on hard sphere model developed previously by Jing and Karato (2011).

(2) On the dihedral angle

The dihedral angle is an important parameter to control the velocity of seismic waves in a partially molten material. However, a key step in determining the dihedral angle is to make sure that a sample reaches textural equilibrium (at least approximately). In some cases, it takes long time to reach equilibrium, and if a sample texture (melt geometry) is not equilibrium, then the applicability of such results is dubious. This issue has been recognized at least for 35 years from the beginning of such a study by Harve Waff (in late 1970s). For instance, [Waff and Blau, 1982] annealed their samples to 720 hours, and showed the evolution of apparent dihedral angle up to ~200 hours. Obviously the time needed for near equilibrium geometry depends strongly on grain-size, but it is essential to confirm near equilibrium geometry. The authors should show evidence for near equilibrium geometry of melt.

7. We strongly agree with the reviewer that the wetting properties of melt are critical for the propagation of seismic waves. We believe that in the presence of hydrous melt the kinetics of equilibration are very fast. We kept the molten samples in the same conditions for up to an hour to observe the effect of textural maturity. Based on steady sound wave velocity values together with melt texture information such as near zero dihedral angle ($\theta < 10^\circ$), extensive wetting of crystal faces and smoothly curved solid melt interfaces in our samples, we conclude that our partially molten samples are in a near-equilibrium state. (L 38-51 and L 101-108)

(3) On the velocity measurements and their interpretation

They measured seismic wave velocities as a function of melt fraction. I find two major problems in this part (in addition to the issue of textural equilibrium discussed above).

(3-1) The first problem is related to the important distinction between the degree of melting and the melt fraction (the degree of melting and the melt fraction are different in the open system, and it is the melt fraction and not the degree of melting that affects seismic wave velocity). When one wants to show the influence of melt fraction on seismic wave velocity, it is better to prepare such a diagram for fixed (but different) melt composition. In contrast, if one conducts an experimental study in a closed system (most of lab studies are for a closed system) where melt fraction agrees with the degree of melting, then as melt fraction changes the melt composition also changes. Consequently, the application of such results will be complicated. There is no discussion on this issue.

8. It is completely true that in a closed system when the melt fraction increases, due to temperature for example, the melt composition evolves, hence the dihedral changes and consequently the velocity. In this sense, the experiments reproduce partial melting in the mantle. One could do experiments by varying the degree of melting using the same incipient

melt composition (similar to Chantel et al. 2016), as suggested by the comment, but this cannot represent the melting in the mantle. This is not like in the simple case of olivine + MORB (Chantel et al. 2016), where the melt fraction can be varied without substantially changing the melt composition. Indeed, a higher melt fraction of incipient melt is impossible to create in nature because if a high melt fraction occurs, it means that the melt has percolated from somewhere else and accumulated, but at the same time it should also re-equilibrate, and only once it has created channels can it move without re-equilibrating (as is seen for mid-oceanic ridge basalts).

Thus in our experiments, the average degree of melting and the melt fraction are equal. Accordingly, Fig. 4 in our revised manuscript illustrates the V_s velocity drop for a given melt fraction. During melting of peridotite, melt composition changes in response to the degree of melting. So the measured seismic velocity should be affected by the subtle variations associated with the change in melt chemistry. However, for small melt fractions (less than 1 % in this case) the chemical variations are so minimal that their influence on the seismic velocity can be ignored (e.g. Hier-Majumder et al. 2014, Afonso & Schutt 2012).

(L 126-128)

(3-2)The second is the interpretation of their results in terms of a model. The influence of partial melting on seismic wave velocities has been studied by many mostly theoretically (e.g., [O'Connell and Budianski, 1974; 1977; Stocker and Gordon, 1975; Takei, 2002]), but [Sato et al., 1989] published a paper on the experimental results at low P (0.5-1.0 GPa). These theoretical papers predict a relation between velocity reductions and melt fraction that depends on the wetting behavior (dihedral angle). ~4 % velocity reduction for ~0.7% melt would correspond to some dihedral angle. How does this result compare with those by Sato et al. (1989)? What dihedral angle do their results correspond to? Does it agree with [Yoshino et al., 2007]'s results? With this dihedral angle, would melt stay there or should it move down or up (what about compaction)? These discussions are important but I do not see any of them.

9. A detailed description of the dihedral angles is included in the revised manuscript (L 95-108). We have also prepared a new figure (Figure 3) illustrating the change of dihedral angle in our samples with melt fraction. For all melt fractions the measured dihedral angle is lower than 10° , while the inset shows that for 0.7 % melt fraction the value of dihedral angle should be below 5° . Our results agree well with Yoshino et al. 2007, which predicts dihedral angles close to 0 for hydrous melt at high pressure. The dihedral angle plays an important role in modifying the seismic wave velocity. Melts with low dihedral angles are expected to facilitate propagation of seismic waves through well wetted grain boundaries.

Sato et al. 1989 is the first experimental study on seismic velocity of partial molten peridotite. While they report reduced seismic velocity upon melting, the melt fraction has to be more than 2 % in order to observe a rapid velocity drop. We believe that this inconsistency may stem from the different wetting properties of dry and hydrous melts, as volatiles, particularly H_2O , are known to improve the wetting properties.

We have also made a comparison between experimental and theoretical investigation into the influence of partial melting on seismic velocity. We found that theoretical models are less sensitive to melt fraction. For example, the calculations based on finite element method on melt geometries led [Hammond and Humphreys, 2000] to conclude that more than 1 % melt is required to explain 3.6 % and 7.9 % velocity reduction for Vp and Vs respectively, while in the laboratory we observe 6.2 % and 11 % Vp and Vs reductions [Chantel et al., 2016]. The naturally occurring, randomly distributed melt [Sato et al., 1989; Faul et al., 2004; Chantel et al., 2016] showed to have a significant effect on seismic velocity compared to the simplified melt textures assumed in theoretical models [Hammond and Humphreys, 2000; Takei, 2002; Yoshino et al., 2005] or analogue systems [Takei, 2000].

To summarize, the following compares 3 different scenarios, purely theoretical (Takei 2000), theory mixed with experimental observations (Yoshino et al. 2005) and experimental (Chantel et al. 2016).

The considerably high melt volume fractions required in theoretical models can be attributed to the idealized geometries, such as plane cracks, spheres, ellipsoids, or simplified cusped forms, which may not represent the true melt geometries in naturally occurring melt [Kohlstedt, 1992; Faul et al., 1994].

In this study, the calculated Poisson's ratio is about 0.31, close to the 0.29-0.30 of mantle values. Based on this value, strong compaction due to presence of melt may not be expected at the LVL. This may support the melt entrainment process (Karato et al. 2006) as an appropriate mechanism to remove excess melt from the LVL.

(4) On the influence of temperature on the melt density

On page 4 (lines 103-105), the authors present a discussion on the influence of temperature on melt density. They argue that at relatively hot mantle, relatively low melt density because the degree of melting is high and less FeO. But how about the influence of water content? At

high temperature, the water content in the melt will be smaller making the melt density to be higher. One needs to compare the influence of both FeO and H₂O.

10. Omitting the effect of water in this particular discussion was an error, and we thank the reviewer for pointing this out. In the revised version we include a discussion on the effect of water. (L 157-159)

(5) On the conversion of the melt fraction to the water content in the transition zone
The authors use their results on the necessary melt fraction to explain the reduction of shear wave velocity to infer the water content in the transition zone. There is an important jump in the logic here. The magnitude of velocity reduction is determined by the melt fraction. The water content in the transition zone will determine the degree of partial melting. This is not the same as the fraction of melt. These two agree only when melt stays the place where it is formed (batch melting). A discussion is necessary to support their conclusion on the water content in the transition zone from the inferred melt fraction in a layer above the 410-km.

11. We have addressed this question in your general comments section.

In addition, there are many grammatical errors. I corrected some, but an editorial office must take a look at this issue.

12. Thank you. The revised version has been language edited by a native-English expert.

I believe that this paper contains a very useful data set, but the interpretations contain several fundamental issues. I hope that the authors can address the comments/questions listed above.

13. Thank you. The revised version has benefitted from your constructive reviews. We hope our responses and subsequent modifications to the manuscript address these concerns.

Shun-ichiro Karato

REVIEWERS' COMMENTS:

Reviewer #1 (Remarks to the Author):

The manuscript improved as a result of the authors' response to the reviews. Most reviews were well addressed and the revised manuscript reads more clearly than the original. The results are impressive and a broad audience will be interested in the potential implications for mantle water storage so I think the manuscript is good candidate for publication. One small but important remaining concern that I have is that the manuscript may give the false impression that the LVL is present over 100% of the surface of the 410, which is not justified based on the existing literature. A few more detailed comments are included below.

It should be noted somewhere in the text that the LVL is sporadically observed across the globe, but it is not present everywhere so the term 'global feature' (line 26) requires just a bit of further explanation. Reference 5 does not claim that the LVL is present everywhere as it shows many locations where the LVL is not detected in addition to positive detections and ambiguous areas. This is similar to other subsequent converted and reflected wave studies (e.g., Schmandt et al., 2011; Wei and Shearer, 2017). It would be productive to avoid giving the false impression that the seismic literature indicates that the LVL exists everywhere, especially because many readers of this manuscript may not be very familiar with the seismic literature.

Given the heterogeneous nature of the LVL, the authors should specify in the 'Water content...' section if their estimate of average MTZ H₂O content depends on the total area of the LVL, or just the mean V_s decrease of the sporadically present LVL from which they infer melt fraction. Aside from clarifying the assumptions of the estimated H₂O content for readers, this could be useful for future studies if constraining variations in the amount of melt could provide insight into any large-scale variations in MTZ H₂O storage.

It is very helpful that the authors provided more thorough information on the effects of volatiles other than H, variable Fe content, and dihedral angle for relevant melt fractions.

Response to reviewers' comments

We thank reviewer #1 for his/her in-depth reviews, which improved the quality of the manuscript. The point-by-point responses to reviewer's comments follow in blue font. The modifications to the MS texts are shown in red.

Reviewer #1 (Remarks to the Author) :

The manuscript improved as a result of the authors' response to the reviews. Most reviews were well addressed and the revised manuscript reads more clearly than the original. The results are impressive and a broad audience will be interested in the potential implications for mantle water storage so I think the manuscript is good candidate for publication.

Thank you for your constructive comments on this study.

One small but important remaining concern that I have is that the manuscript may give the false impression that the LVL is present over 100% of the surface of the 410, which is not justified based on the existing literature. A few more detailed comments are included below.

It should be noted somewhere in the text that the LVL is sporadically observed across the globe, but it is not present everywhere so the term 'global feature' (line 26) requires just a bit of further explanation. Reference 5 does not claim that the LVL is present everywhere as it shows many locations where the LVL is not detected in addition to positive detections and ambiguous areas. This is similar to other subsequent converted and reflected wave studies (e.g., Schmandt et al., 2011; Wei and Shearer, 2017). It would be productive to avoid giving the false impression that the seismic literature indicates that the LVL exists everywhere, especially because many readers of this manuscript may not be very familiar with the seismic literature.

We agree with the reviewer. The LVL is a widespread feature but based on available seismic data, we cannot confirm that the LVL is present over 100% of the surface of 410-km discontinuity. In the revised manuscript, we have modified our original sentence to clarify this point.

(L-27-28) "The LVL is reported to be a widespread seismic anomaly⁵, with an average 4 % Vs velocity drop across a narrow, ~ 50-60-km-thick region atop the mantle transition zone MTZ."

Given the heterogeneous nature of the LVL, the authors should specify in the 'Water content...' section if their estimate of average MTZ H₂O content depends on the total area of the LVL, or just the mean Vs decrease of the sporadically present LVL from which they infer melt fraction. Aside from clarifying the assumptions of the estimated H₂O content for readers, this could be useful for future studies if constraining variations in the amount of melt could provide insight into any large-scale variations in MTZ H₂O storage.

The water content in the MTZ was calculated using the melt fraction, consistent with our velocity model. We have already mentioned these assumptions in our manuscript (L 170-175).

We agree that the second point raised by the reviewer is not clearly addressed in our discussion and we have included a sentence clarifying this issue.

(L 176-177) However, variation of melt fraction within the LVL may indicate regional variations of water content in the MTZ.

It is very helpful that the authors provided more thorough information on the effects of volatiles other than H, variable Fe content, and dihedral angle for relevant melt fractions.

As per reviewer's suggestion, we have modified the supplementary table 1, in order to indicate the melt chemistry (including volatile contents) and corresponding dihedral angle for each melt fraction. (Supplementary Table 1)

Please note that our starting peridotite composition is free of CO₂. Raman analyses of melt after high pressure-temperature experiments indicate the CO₂ contents are below detection limits, indicating CO₂ adsorption during the experiment is less significant.